# Identification and Characterization of Fibronectin-Binding Peptides in Gelatin

**DOI:** 10.3390/polym14183757

**Published:** 2022-09-08

**Authors:** Yuying Liu, Jianping Gao, Lin Liu, Jiyao Kang, Xi Luo, Yingjun Kong, Guifeng Zhang

**Affiliations:** 1State Key Laboratory of Biochemical Engineering, Institute of Process Engineering, Chinese Academy of Sciences, Beijing 100190, China; 2School of Chemical and Engineering, University of Chinese Academy of Sciences, Beijing 100049, China; 3Library of Shandong Agricultural University, Tai’an 271018, China

**Keywords:** collagen, gelatin, fibronectin-binding, peptides, HPLC-MS

## Abstract

Collagen and fibronectin (FN) are important components in the extracellular matrix (ECM). Collagen-FN binding belongs to protein-protein interaction and plays a key role in regulating cell behaviors. In this study, FN-binding peptides were isolated from gelatin (degraded collagen) using affinity chromatography, and the amino acid sequences were determined using HPLC-MS. The results indicated that all FN-binding peptides contained GPAG or GPPG. Matrix-assisted laser desorption/ionization time-of-flight mass spectrometry (MALDI-TOF MS) and dual-polarization interferometry (DPI) were used to analyze the effects of hydroxylation polypeptide on FN binding activity. DPI analysis indicated that peptides with molecular weight (MW) between 2 kDa and 30 kDa showed higher FN-binding activity, indicating MW range played an important role in the interaction between FN and peptides. Finally, two peptides with similar sequences except for hydroxylation of prolines were synthesized. The FN-binding properties of the synthesized peptides were determined by MALDI-TOF MS. For peptide, GAPGADGP*AGAPGTP*GPQGIAGQR, hydroxylation of P8 and P15 is necessary for FN-binding. For peptide, GPPGPMGPPGLAGPPGESGR, the FN-binding process is independent of proline hydroxylation. Thus, FN-binding properties are proline-hydroxylation dependent.

## 1. Introduction

Collagen and fibronectin (FN) play a vital role in the extracellular matrix (ECM). Collagen is a biopolymer synthesized by animal cells, due to its special physiological function and high nutritional value, it is widely used in healthcare, cosmetics, and medical materials. It could prevent cardiovascular and cerebrovascular diseases and can lower blood pressure [1,2,3]. Collagen could interact with FN [4], but gelatin and denatured collagen have stronger binding activities than collagen [5]. In the ECM, the conjugation has important biological significance. The binding of collagen to FN could cause a conformational change in FN, leading to various biological responses, such as attachment to the cell surface to promote cell migration and wound repair, as well as differences in integrin receptor binding [6]. However, the absence of collagen in cultured primary fibroblasts will affect the arrangement of FN in the ECM [7]. Consequently, the study of collagen and FN binding sites is key to some relevant biological interactions.

FN, a 450 kDa glycoprotein, is composed of three modules (F1, F2, and F3). In the binding of FN and collagen, it is known that the 42 kDa fragment near the N-terminal of FN, 6F1-1F2-2F2-7F1-8F1-9F1 (n denotes the nth type in the nomenclature nFX) [8], is the collagen-binding domain. The F2 module plays a pivotal role in the interaction, but almost all the modules in the 42-kDa fragment are important [9].

In collagen, the specific FN-binding sites have not yet been identified. The binding activity is influenced by many factors, including the molecular weight (MW) and the hydroxylation modification of collagen. Hydroxyproline is a unique amino acid of collagen, and the content of hydroxyproline differs with age and gender [10,11]. Therefore, hydroxyproline plays a critical role in collagen-FN binding. Due to the stronger binding activity of gelatin, undoubtedly, MW contributes to the binding because the modest MW radically exposed more FN-binding sites in denatured collagen than in collagen. However, these factors have yet to be thoroughly studied. In 1978, Kleinman discovered α1-CB7, digested by CNBr and purified by ion exchange chromatography, in residues 757–791, and the author identified this as the classic FN-binding region [12]. In Dessau’s study, FN could bind to collagen type I, II, III, IV, and V in vitro, as there is a binding site in every helix of collagen. Positions 643–819 in the α1 chain contain an FN-binding site, and a domain homologous to the binding site in residues 693–1101 of the α2 chain was identified by fluorescence chromatography [13]. Gao separated two non-triple helical regions of collagen by FN-sepharose affinity chromatography, size exclusion high-performance liquid chromatography (HPLC), and enzyme-linked immunosorbent assay (ELISA) [14]. In addition, CNBr cleaved collagen α1(II) to obtain CB10 (position 550–900), CB11 (100–400), CB8 (400–550), CB9.7 (900–1000), and CB12 (0–100). In these fragments, CB10 and CB12 had binding activity, as detected by fluorescence polarization, and the activity of CB8 was weaker. Furthermore, α2(I)-CB4 and CB5, which were homologous to CB10 and CB12, respectively, were active in FN binding [15]. More recent research suggested that there were at least fourteen distinct sites in collagen type I to interact with FN: five on each of the α1 chains and four on the α2 chain, using fluorescence anisotropy [16]. However, all the above-mentioned studies used CNBr to cleave collagen and analyzed the binding activity of the cleaved fragments to FN, without specific binding site analysis. These fragments comprised a longer region, and the MW range was broad (from several thousand to tens of thousands). The binding sites were still uncertain. Furthermore, the detection and analysis methods almost used fluorescence chromatography.

High-performance liquid chromatography/tandem mass spectrometry (HPLC-MS) has been successfully applied to the analysis of the digested peptides in collagens or post-translational modification residues of peptides. Dual-polarization interferometry (DPI) has been employed for the analysis of protein-protein/small molecule interactions, protein conformational changes in the solid interface, and protein-solid interface interactions [17]. In this paper, the interaction of peptides and FN was analyzed by HPLC-MS/MS in combination with DPI, and key factors of the binding were investigated. Specifically, bovine gelatin was digested by trypsin after passing through the FN affinity column. The sequence of the separated peptides was analyzed, and the hydroxylation modification position of proline was determined by HPLC-MS/MS. Then, the sequences were synthesized, including hydroxylation modification, and the interaction with FN was detected via MALDI-TOF MS. In addition, the effect of the MW was analyzed by DPI.

## 2. Materials and Methods

### 2.1. Materials

Bovine gelatin (G9382, type B) and human FN were purchased from Sigma-Aldrich (St. Louis, MO, USA). Trypsin (sequencing grade) was obtained from Promega (Madison, WI, USA). Trifluoroacetic acid (TFA) and acetonitrile (ACN) were purchased from Fisher Scientific (Fair Lawn, JN, USA). α-Cyano-4-hydroxycinnamic acid (CHCA) was purchased from Bruker (Billerica, MA, USA). 3-Aminopropyltriethoxysilane (APTES) was purchased from Alfa Aesar (Haverhill, MA, USA). An unmodified Anachip was purchased from Farfield (Sensors Ltd., Salford, UK). The four synthesized peptides were purified by Beijing Scilight Biotechnology Led. Co. (Beijing, China).

### 2.2. Methods

#### 2.2.1. Preparation of FN Sepharose Affinity Media

Sepharose 4FF (National Engineering Research Center for Biotechnology, Beijing, China) was activated by 1,4-butanediol diglycidyl ether. Human FN (1 mg/mL) was coupled to 3 mL of activated sepharose for 24 h at 37 °C. Active groups were blocked with ethanolamine for 4 h at 37 °C. Then, the sepharose was washed alternatively with buffer solutions at pH 8.0 and pH 4.0. The matrix was filled in the column (1.0 cm × 4.0 cm) to prepare FN-affinity column. The same method was also used to prepare the micro-FN-affinity column (100 μL).

#### 2.2.2. Tryptic Digestion

Twenty milligrams of bovine gelatin were dissolved in 5 mL of ultrapure water. Two times the volume of ethanol was added, and the mixture was kept at 4 °C for 30 min. The gelatin solution was centrifuged at 10,000 r/min for 10 min to precipitate the gelatin. The high MW peptides of the precipitate were recovered in trypsin resuspension buffer (pH 8.0), and 1:50 (*w*/*w*) trypsin solution (1 μg/μL in trypsin resuspension buffer, pH 8.0) was added. The mixture was incubated at 37 °C for 12 h.

#### 2.2.3. FN Affinity Chromatography [9,14,17]

The digest was loaded onto the FN-affinity column and stabilized in 20 mM PBS equilibrium buffer (pH = 7.4). The column was washed in equilibrium buffer, and peptides were eluted with 3 M urea in equilibrium buffer. The eluted peptides were desalted by Sephadex G-25 (GE Healthcare, Chicago, IL, USA) and concentrated. The obtained peptides were referred to as sample 1.

The undigested gelatin was separated in the same way, and the isolated peptides were digested by trypsin at 37 °C for 12 h. The obtained peptides were referred to as sample 2.

#### 2.2.4. MALDI-TOF MS Analysis

The MW ranges of samples 1 and 2 were determined using MALDI-TOF-MS (autoflex III smartbeam, Bruker, Bremen, Germany) [18]. The samples were mixed in the ratio 1:1 with the α-cyano-4-hydroxycinnamic acid matrix (5 mg/mL in 50% ACN containing 0.1% TFA), and the samples were spotted on the MALDI target. The mass scan range was set from *m*/*z* 600 to 10,000. A mass scan was performed in the positive ion mode and linear detector.

The synthesized peptides were loaded onto the micro-FN-affinity column, followed by the washing solution. The eluted peptides were also analyzed by MALDI-TOF MS.

#### 2.2.5. HPLC-MS Analysis

The separated peptides in samples 1 and 2 were analyzed by HPLC-MS [19]. The HPLC-MS system consisted of an Agilent 1100 HPLC and MS (LCQ DecaXP, Thermo Electron, San Jose, CA, USA). The samples were loaded onto an Agilent Zorbax SB C18 column (2.1 mm × 150 mm, 5 μm). Mobile phase A was water (containing 0.1% TFA), and mobile phase B was ACN (containing 0.1% TFA). The HPLC gradient was 5–40% B from 0 to 50 min; 40–80% B from 50 to 80 min. The flow rate was 0.2 mL/min. The outlet of the column was introduced into the ion source of an electrospray ionization mass spectrometer. The spray voltage was 4.5 kV, and the capillary was kept at 300 °C. The data acquisition consisted of three scan events: an MS scan, followed by one zoom scan to determine the charge state of the ion, and an MS/MS scan to generate an MS/MS spectrum. The MS scan range was from *m*/*z* 400 to 2000. The zoom scan and tandem mass spectrometry (MS/MS) functions were performed in data-dependent mode. Dynamic exclusion was adopted to one count and a 0.5 min exclusion duration unit. The collision energy value was installed to 35%.

#### 2.2.6. Database Searching and Data Processing

Sequence information from MS/MS data was processed using the Turbo SEQUEST algorithm in Bioworks 3.2 software (Thermo Electron, San Jose, CA, USA). A database was created by extracting bovine type I collagen entries from the Swiss-Prot/TrEMBL database (http://www.expasy.org, accessed on 15 July 2020). The database searches and SEQUEST criteria were based on a previously published method [19].

#### 2.2.7. Surface Modification of DPI Chip

The unmodified DPI sensor chip (AnaChip^TM^, Farfield Sensors Ltd., Manchester, UK) was successively cleaned by ultrasonication in ethanol, acetone, and ultrapure water for 5 min. Then, the chip was immersed in piranha solution (7:3 sulfuric acid to hydrogen peroxide) at 90 °C for 2 h, rinsed in ultrapure water, and dried. APTES was added to anhydrous toluene to create solution concentrations of 0.001 mol/L APTES. The chip was placed in the APTES solution to react for 30 min, followed by ultrasonication in anhydrous toluene. The chip was dried by nitrogen stream. Finally, an amine-functionalized surface formed on the chip [20].

#### 2.2.8. DPI Analysis and Data Processing [17,20]

The modified chip was installed in an AnaLight^®^ Bio200 dual-polarization interferometer (Farfield Sensors Ltd., Salford, UK). The laser beam (λ = 632.8 nm) with two orthogonal polarizations passed through the chip. When the protein was loaded on the chip surface, the upper sensing waveguide was changed, and the lower reference waveguide of the chip was of no influence, causing the laser to produce an evanescent field change. This resulted in a shift of the interference fringes. Subsequently, the change in transverse magnetic (TM) and transverse electric (TE) phases was used to calculate the refractive index and thickness, respectively, via the Maxwell equation. Tris-HCl buffer (pH 7.4, including the 0.1 M NaCl) flowed over the chip surface. The instrument was calibrated with ethanol and H_2_O prior to the experiment. To prepare the chip, 100 μg/mL FN was injected, followed by the sealing agent. Then, the peptides samples were injected, and the surface dimensions and densities were measured. The data were automatically analyzed by AnaLight DAQ.

## 3. Results and Discussion

### 3.1. Separation of Peptides through FN Affinity Chromatography

Due to the affinity of FN with gelatin, peptides in gelatin were separated by affinity chromatography, in which affinity media was prepared by coupling FN on agarose media. The sample was precipitated by ethanol and then was dissolved with PBS buffer (pH = 7.4) before adding to the separating column. The affinity chromatography results were shown in Figure 1. Fragments of nonspecific adsorption were removed by washing the column with 20 mM PBS buffer (pH 7.4), and the binding peptides were eluted with 3 M urea [21]. The elution peak with the high concentration of salt would affect the mass spectrometric detection. The peptide fraction was desalted with SephadexG-25(10 mL) (Figure 2). Figure 1 illustrated that not all the peptides from the digested gelatin could bind to the FN, and only a few containing special sequences could bind to the N-terminal 42 kDa region of FN. We inferred that the special sequences may play an important role in FN-affinity. In Figure 2, it could be seen that the MW range of the separated peptides changed. The main peaks were distributed in 6–8 mL, which were some peptides with larger MW.

### 3.2. MALDI-TOF MS Analysis of the MW Range

The MW range of the two separated samples was analyzed using MALDI-TOF MS. The mass spectra in Figure 3 and Figure 4 showed that the MW range of both samples was from approximately 1000 to less than 6000 Da but was mainly distributed between 2000 and 3000 Da. In Figure 3, the analysis for sample 1 indicated that the separated peptides were degraded into the lower MW range, and the sequences could be analyzed by HPLC-MS. The analysis of sample 2 in Figure 4 indicated that the lower MW peptides of the digested gelatin could also bind to FN alone.

### 3.3. HPLC-MS Analysis of the Peptide Sequences

Bovine gelatin, the separated gelatin (not digested), sample 1, and sample 2 were analyzed by HPLC-MS, and the corresponding chromatograms are shown in Figure 5. The similarity of the traces in Figure 5A,B indicated that gelatin contains many peptides that could bind to FN. In contrast to Figure 5B–D, the retention time in Figure 5C,D was much shorter than that in Figure 5B, further indicating that the larger peptides in the gelatin were degraded into smaller peptides. Database searching and software analysis (2.2.6) of the mass spectra shown in Figure 5C,D were performed, and the results are listed in Table 1. Thirteen varieties of peptides were found in both samples 1 and 2. Sample 1 was obtained by tryptic digestion of gelatin followed by FN affinity column separation, while sample 2 was obtained by FN affinity column separation of gelation followed by tryptic digestion. The identified peptides in sample 2 consisted of shorter sequences that could directly bind to FN, and so do the extended sequences before tryptic digestion. Therefore, the region of the FN-binding sites could be narrowed to these sequences, including GPAG and GPPG, which were repeated in one sequence. The hydroxylation modification degree of proline in the repeated GPAG and GPPG sequences was not high. In the reported literature, the Y position of G-X-Y in the FN-binding region had lower hydroxyproline content and stronger hydrophobicity, causing the collagen helix expansion to expose the active FN-binding site [22]. Consequently, the active sites of FN-binding might exist in GPAG and GPPG. Ingham K.C. found that type I collagen contains at least 14 cryptic FN binding sites of similar affinity [16].

Then, the sequence RGEP*GPP*GPAG (*indicated the hydroxylation modification of the proline) and GAPG (n = 39, n represents the number of repeats in the chain), GPAG (n = 61), GARG (n = 19), and GERG (n =19) were synthesized, but they failed to bind FN. The moderate hydroxylation modification of proline might play an important role in recognition by FN. Furthermore, the shorter synthesized peptides, which contained FN-binding sites, failed to bind FN, indicating that the combined action of other residues was clearly required for full affinity.

The specific “GPPG” sequences in different peptides are highlighted in red. The specific “GPAG” sequences in different peptides are marked with underlines.

In the study by Kleinman H. K., α1-CB7, a type I collagen, was digested with cyanogen bromide and was identified as the classic fragment of the FN-binding [12]. The fragment lies within residues 757–791 of the α1 chain and contains the vertebrate collagenase (MMP-1) cleavage site of residues 775–776. In our research, *m*/*z* 1043.31 in Table 1 was doubly charged in the zoom scan, and the ion corresponded to the sequence, GAPGADGP*AGAPGTP*GPQGIAGQR, in the region of α1.757–780. The sequence belonged to α1-CB7 and contained the cleavage site Q-G#I-A (# represents the bond cleaved) of MMP-1 [23]. This suggested that the fragment, which independently interacted with FN in the domain of α1.757–791, could reduce to the sequence, GPQGIAGQR, in the region of α1.772–780. In addition, the sequence, GP*AG, might be another binding site. The sequence GLP*GVAGSVGEPGP*LGIAGPPGAR in the region of α2.760–783 corresponded to *m*/*z* 1058.07 (doubly charged) in Table 1, and it contained the cleavage site L-G#I-A. As a result, we inferred that in addition to the collagenase cleavage site, GPAG and GPPG were also binding sites of FN, but the excessive hydroxylation modification of the proline in the sequence GPPG weakened the adhesion to FN.

Studies showed that both α1.552–819 and 643–932 exhibit high FN-binding activity [13]. In Table 1, the peptides numbered 1–4 belonged to the region (Figure 6). The retention time of *m*/*z* 796.8, 1042.3, 908.9, and 780.9 with two charges was 29.46, 38.34, 30.25, and 30.18 min, respectively. In addition, *m*/*z* 793, a double charge, originated from the sequence GANGAP*GIAGAP*GFP*GAR in Figure 7. The result was consistent with that reported by Fietzek et al., in which α1-CB8 and α1.124–402 could bind to FN [24]. The region of α1.220–237 shortened the binding region of α1-CB8 and made the binding site more obvious.

Our research identified some binding peptides from the α2 chain. Guidry C. digested collagen I and II using CNBr and found that both CB4 from α2(I) homologous with CB10 from α1(II) and CB5 from α2(I) homologous with CB12 from α1(II) exhibited FN-binding activity. CB4 and CB5 were located in the regions of α2(I).0–300 and 693–1011, but the binding activity of CB4 was weaker than that of CB5 [15]. Table 1 showed that the peptides of regions α2.102–111, α2.708–723, α2.856–873, and α2.931–941 were included in the binding region. This further suggested that the bound region could cut down these sequences. They contained the same peptide sequences of GPAG and GPPG. Therefore, GPAG or GPPG was the binding site of FN. Moreover, this is the first report on these sequences in α1.91–99, α2.328–341, α2.451–465, α2.469–486, and α2.672–692 (Table 1) and their binding with FN. For example, the sequence GFSGLDGAK corresponded to *m*/*z* 851.4, and the matched b and y ions were shown in Figure 8.

### 3.4. DPI Analysis of the Effect of MW on FN Binding

After FN was immobilized on the chip, the 1, 3, and 10 kDa peptides and bovine gelatin were, respectively, injected into the two channels of DPI and reacted with immobilized FN. The MW range of bovine gelatin was 1.4–33 kDa, as determined by gel filtration chromatography. The adsorption conditions of these samples were shown in Figure 9, Figure 10 and Figure 11, including the real-time profile (A) and the resolved layer thickness-mass (B).

In Figure 9B, the thickness increased from 11.60 nm to 12.55 nm, and the mass increased from 3.68 to 3.86 ng/mm^2^. The obvious increases in thickness and mass demonstrated that gelatin of MW range 1.4–33 kDa interacted with the immobilized FN. Certainly, not the entirety of the MW range (1.4–33 kDa) could bind to FN; thus, 1, 3, and 10 kDa peptides were further analyzed.

According to DPI data analysis, the 1 kDa sample failed to bind FN (Figure not shown). Although the thickness did not increase significantly, there was a significant increase in the mass in the 3 kDa sample (Figure 10). This proved that the 3 kDa sample bound to FN. Similarly, the mass and thickness of the 10 kDa sample indicated that it could interact with FN (Figure 11).

It is known that the FN-binding activity of collagen is lower than that of gelatin [5]. Our study suggested that the peptides lower than 1 kDa did not readily bind to FN. Thus, the bigger MW of peptides, the better the FN-binding activity. In other words, there was stronger binding activity in the suitable MW range of peptides, because a modest MW might radically expose more FN-binding sites than collagen. In this paper, peptides in the range of 2–30 kDa exhibited better affinity to FN. Therefore, the MW plays an important role in the interaction between FN and peptides.

### 3.5. MALDI-TOF MS Analysis of the Effect of Hydroxylation Modification on FN Binding

The sequence and MW of the four synthesized peptides were listed in Table 2. They were the peptides of FN-binding in Table 1 and contained two non-hydroxylation (1 and 3) and two hydroxylation modifications (2 and 4). In order to investigate the effect of hydroxylation modification on FN binding, peptides 1 and 2, and peptides 3 and 4, were, respectively, mixed in a 1:1 mass ratio and separately named GIA and GPP. The two mixed samples were separately loaded onto the micro affinity column. The column was eluted until the washing solution did not include the sample by MALDI-TOF MS, following the elution solution analyzed. Figure 12 and Figure 13 showed the MALDI-TOF mass spectra of the two samples’ elution.

In Figure 12A, *m*/*z* 2056.01 and 2088.30 were produced by peptides 1 and 2 from Table 2, respectively, and *m*/*z* 2078.19 was *m*/*z* 2056.01 + Na^+^. The figure showed that the washing solution no longer included the GIA sample after the column was fully washed (Figure not shown). By comparing Figure 12A,B, the larger peak area of peptide 1 developed into smaller than that of peptide 2. The differences between the spectra indicated that the hydroxylation modification of peptide 2 led to stronger binding activity than the non-hydroxylation of peptide 1. Again, the appropriate position and amount of hydroxyproline in the sequence of peptide 2 had a significant influence on the binding activity.

In comparison with Figure 13A, the peak areas in Figure 13B did not significantly change. We hypothesized that the amount of proline in the sequence of peptide 3 was so high that slight hydroxylation modifications might influence the binding, or even inhibit it.

Interestingly, all four synthesized peptides could interact with FN, and it was incorrect to infer that only the sequences with hydroxylation modification could bind to FN. The addition of the binding was also attributed to the position and amount of hydroxyproline, both of which are essential.

## 4. Conclusions

Affinity chromatography was used to isolate peptides that could bind to FN, and the amino acid sequence of peptides was determined by HPLC-MS. MALDI-TOF MS and DPI were used to analyze the effects of MW and hydroxylation modification of peptides on FN binding activity.

The sequences of peptides that could bind FN all contained GPAG or GPPG; thus, they may be regarded as the core sequences of FN binding peptides. The MW of these peptides was mainly distributed in the range of 1000–2000 Da, and the FN-binding regions in peptide sequences were identified. These peptides, with MW between 1000 and 2000 Da, are also more easily absorbed by the human body, providing a foundation for the development of new functional health products.

DPI was used to measure the adsorption thickness and mass of peptides with different MW ranges on a chip with immobilized FN. Peptides with MW below 1 kDa did not readily bind to FN, while peptides with MW between 2 and 30 kDa were more likely to bind to FN. The result indicates that peptides with moderate MW exert greater binding activity, where more FN binding sites are exposed in the peptide sequence. Therefore, the MW range plays an important role in the interaction between FN and peptides.

Two different sequences of peptides (including hydroxylated and unmodified peptides) were synthesized, and the amount of hydroxylated and unhydroxylated peptides was mixed. MALDI-TOF MS was used to measure the influence of hydroxylated modification on FN binding, and all four synthetic polypeptides could bind to FN. However, the location and amount of hydroxylation modification play a key role in binding strength.

In this paper, bio-enzymatic hydrolysis technology was used to degrade gelatin, and combined with mass spectrometry analysis, the core sequence of FN binding in peptides was identified at the molecular level. The FN binding domain of the peptide sequence was shortened, providing a strategy for new raw materials and a basis for the study of functional factors of functional food. In this paper, DPI and MALDI-TOF MS were used for the first time to analyze the dynamic binding process of peptides to FN online, and the influence of various factors on the binding was discussed. The results provide a new technical method of analysis, as well as a theoretical basis for studying the binding characteristics of peptides to FN.

## Figures and Tables

**Figure 1 polymers-14-03757-f001:**
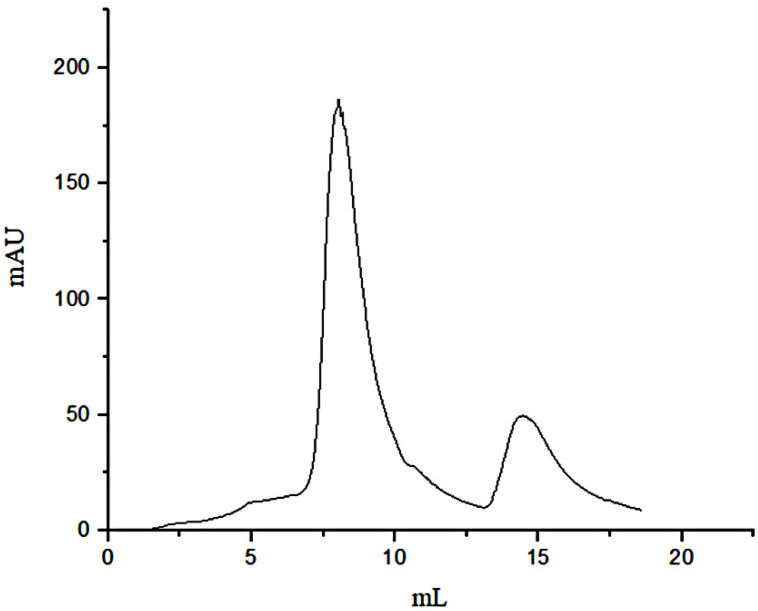
Affinity chromatography of separation peptides in the gelatin.

**Figure 2 polymers-14-03757-f002:**
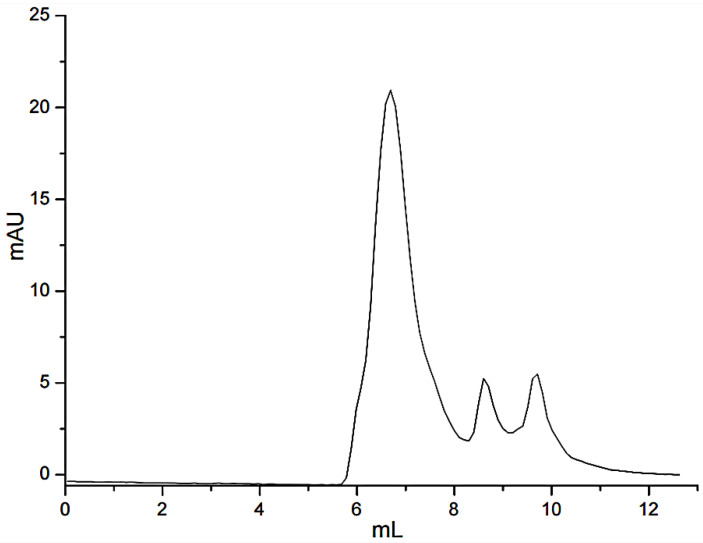
Gel filtration chromatograms of the desalinated peptide fraction.

**Figure 3 polymers-14-03757-f003:**
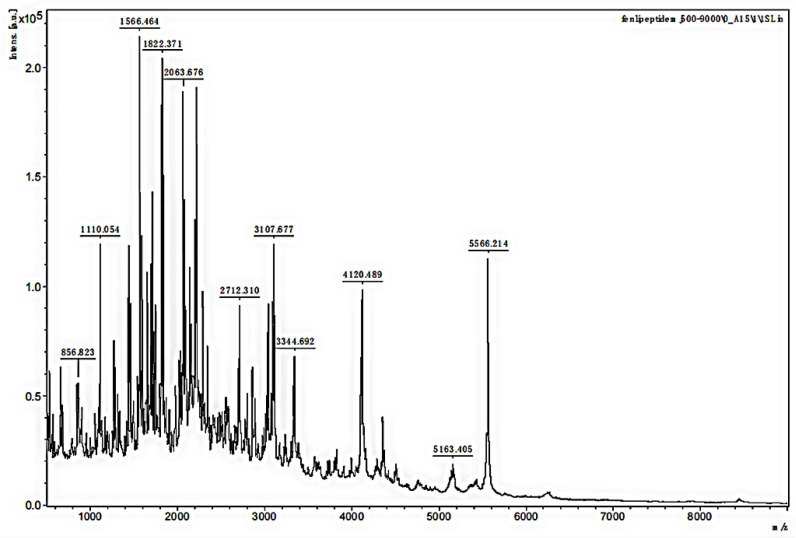
MALDI-TOF MS mass spectra of sample 1.

**Figure 4 polymers-14-03757-f004:**
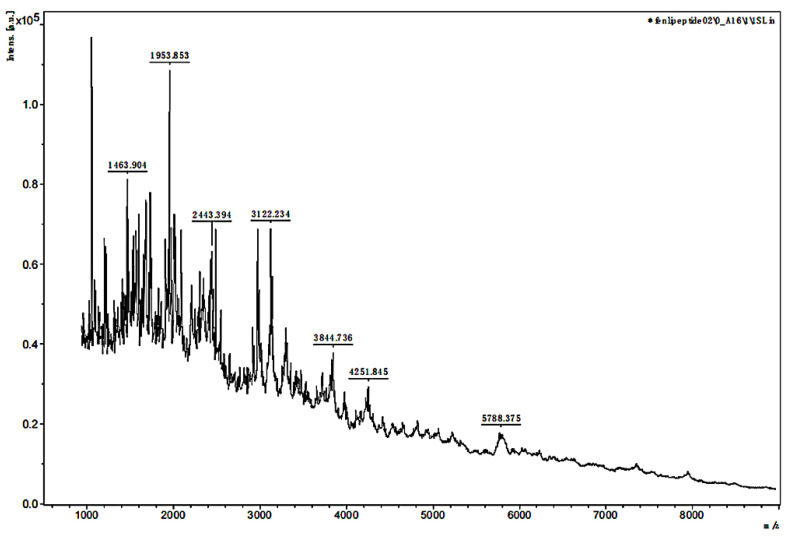
MALDI-TOF MS mass spectra of sample 2.

**Figure 5 polymers-14-03757-f005:**
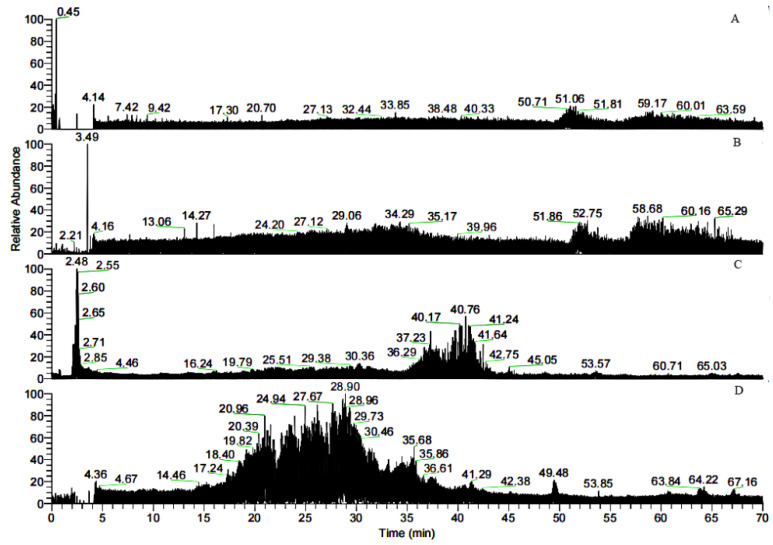
Total ion chromatograms of the bovine gelatins digested at 37 °C for 12 h. (**A**) Bovine gelatin; (**B**) separated gelatin before sample 1 was digested; (**C**) sample 1; (**D**) sample 2.

**Figure 6 polymers-14-03757-f006:**
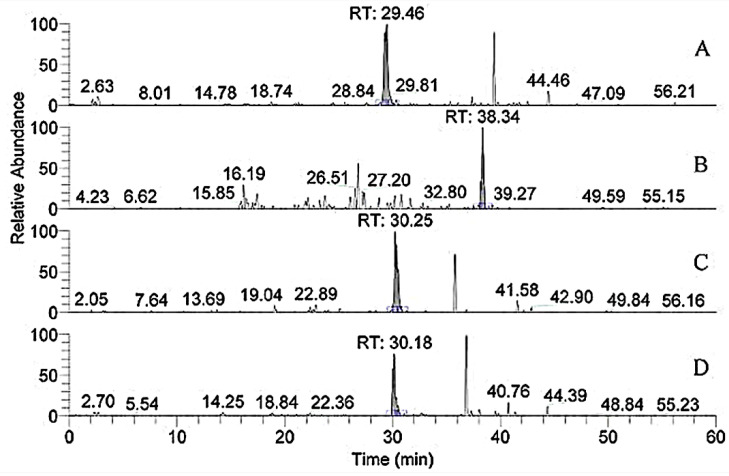
Extracted ion chromatogram of the peptides, numbers 1–4, in Table 1. (**A**) *m*/*z* 796.8; (**B**) *m*/*z* 1042.3; (**C**) *m*/*z* 908.9; (**D**) *m*/*z* 780.9.

**Figure 7 polymers-14-03757-f007:**
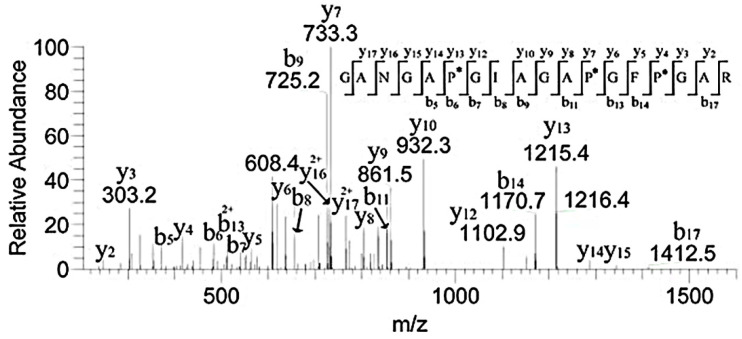
MS/MS spectrum of *m*/*z* 793.6 corresponding to the sequence GANGAP*GIAGAP*GFP*GAR. “*” indicated the hydroxylation modification of the proline.

**Figure 8 polymers-14-03757-f008:**
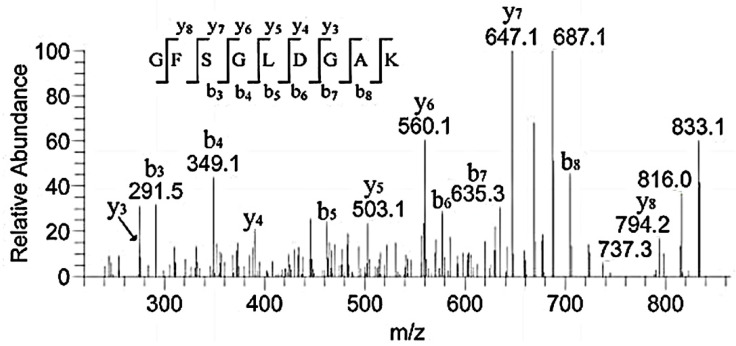
MS/MS spectrum of *m*/*z* 851.4 corresponding to the sequence GFSGLDGAK.

**Figure 9 polymers-14-03757-f009:**
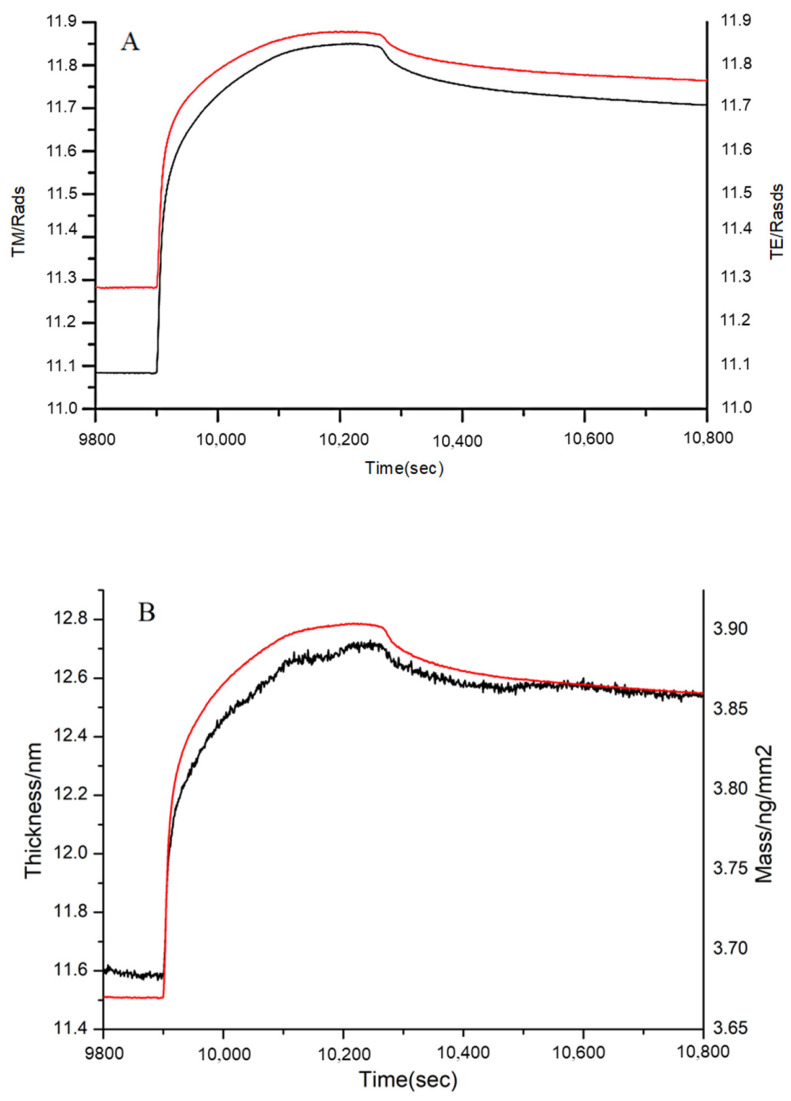
Adsorption of gelatin to immobilized FN on the chip surface as shown by (**A**) the measured TM (black line) and TE (red line) phase changes, and (**B**) the resolved layer thickness (black line) and mass (red line).

**Figure 10 polymers-14-03757-f010:**
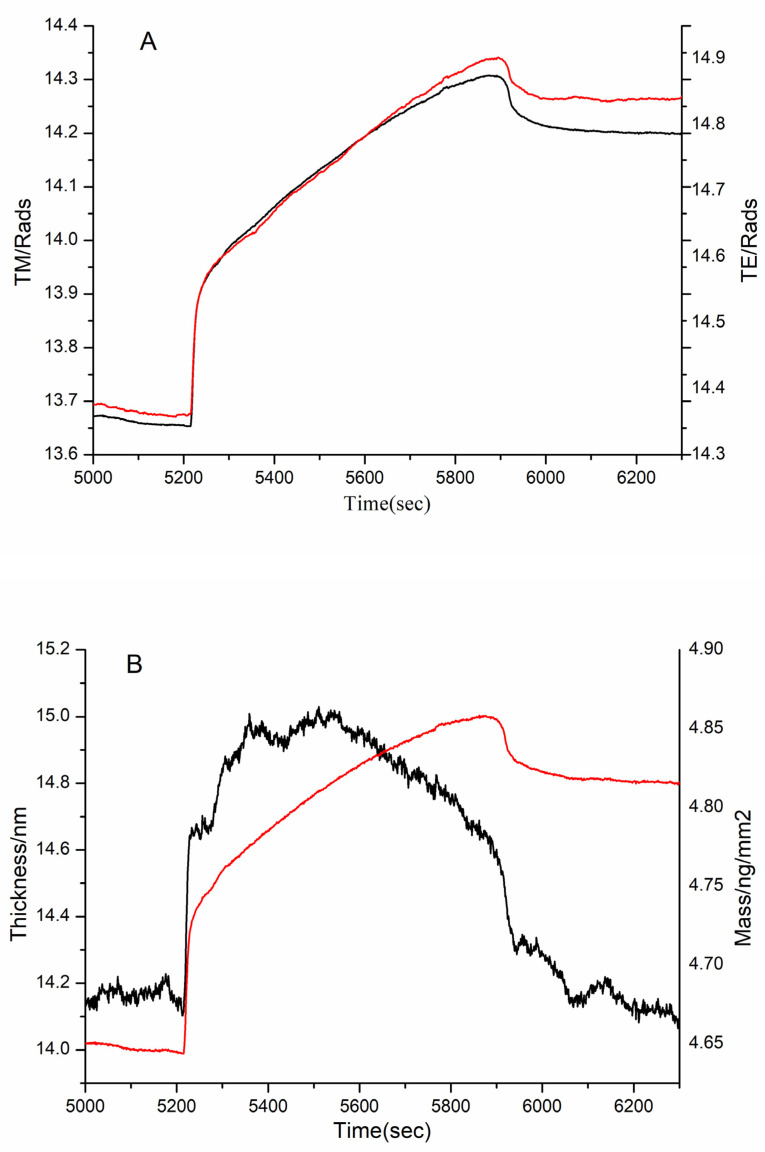
Adsorption of 3 kDa sample to immobilized FN on the chip surface as shown by (**A**) the measured TM (black line) and TE (red line) phase changes and (**B**) the resolved layer thickness (black line) and mass (red line).

**Figure 11 polymers-14-03757-f011:**
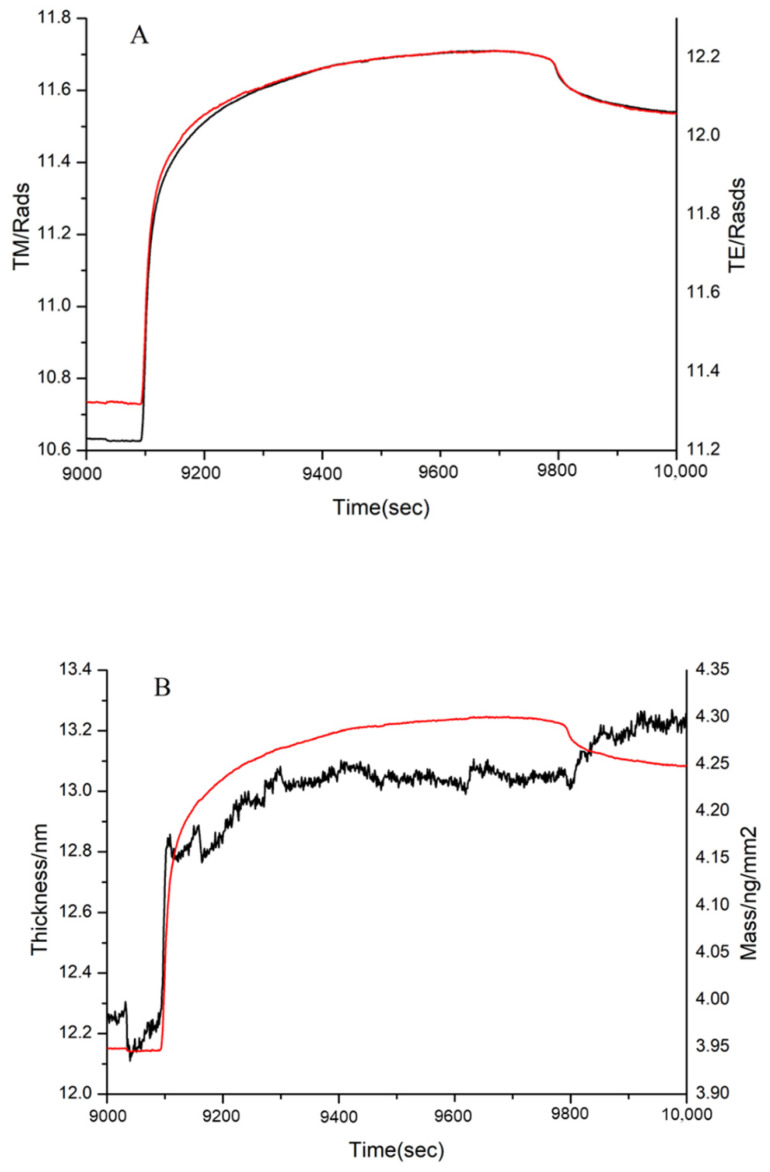
Adsorption of 10 kDa sample to immobilized FN on the chip surface, as shown by (**A**) the measured TM (black line) and TE (red line) phase changes and (**B**) the resolved layer thickness (black line) and mass (red line).

**Figure 12 polymers-14-03757-f012:**
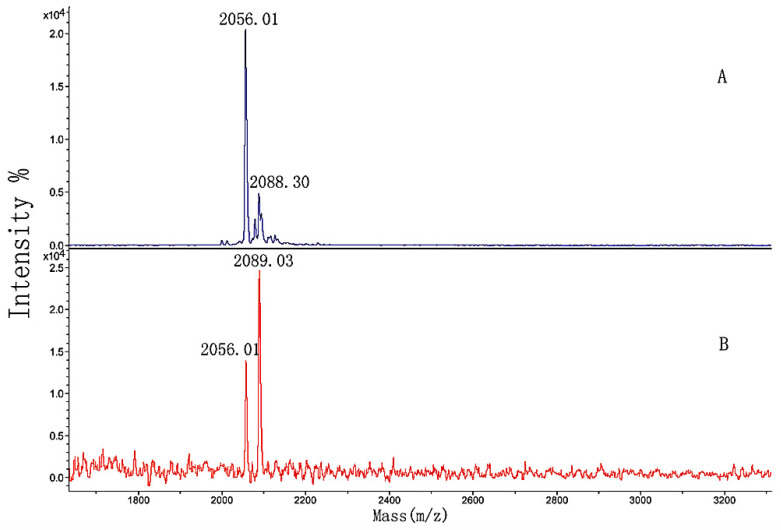
MALDI-TOF MS spectra of (**A**) GIA sample and (**B**) the elution solution.

**Figure 13 polymers-14-03757-f013:**
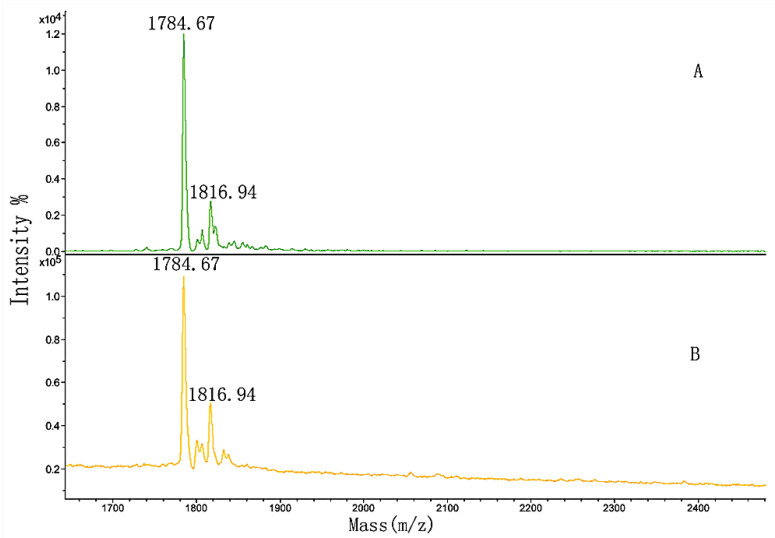
MALDI-TOF MS spectra of (**A**) GPP sample and (**B**) the elution solution.

**Table 1 polymers-14-03757-t001:** Peptides in samples 1 and 2 identified from database searching.

Number	*m*/*z*	Position	Sequence
1	1590.82	α1.586–603	GLTGPIGPP*GPAGAP*GDK
2	2089.02	α1.757–780	GAPGADGP*AGAPGTP*GPQGIAGQR
3	1816.88	α1.817–836	GPP*GPMGPPGLAGPP*GESGR
4	1560.81	α1.889–906	GETGPAGPAGPIGPVGAR
5	868.47	α2.102–111	VGAP*GPAGAR
6	1287.64	α2.328–341	GFP*GSP*GNIGPAGK
7	1427.74	α2.451–465	GIP*GEFGLPGP*AGAR
8	1580.77	α2.469–486	GPP*GESGAAGPTGPIGSR
9	1845.92	α2.672–692	TGPP*GP*SGISGPP*GPPGP*AGK
10	1492.70	α2.708–723	SGETGASGPP*GFVGEK
11	2115.13	α2.760–783	GLP*GVAGSVGEPGP*LGIAGPPGAR
12	1580.79	α2.856–873	GEP*GP*AGAVGP*AGAVGP*R
13	895.46	α2.931–941	GPAGPSGPAGK

**Table 2 polymers-14-03757-t002:** Sequence and molecular weight of the four synthesized peptides.

No.	Mixed Sample Name	No hyp Modification	Molecular Weight
1	GIA	GAPGADGPAGAPGTPGPQGIAGQR	2058.08
2		GAPGADGP*AGAPGTP*GPQGIAGQR	2089.47
3	GPP	GPPGPMGPPGLAGPPGESGR	1784.53
4		GPP*GPMGPPGLAGPP*GESGR	1816.36

## Data Availability

The data presented in this study are available on request from the corresponding author.

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
