# Peer review of "Identification and Characterization of Fibronectin-Binding Peptides in Gelatin"

_polymers, 2022, doi:10.3390/polym14183757_

Round 1
Reviewer 1 Report
The paper studies collagen peptides that could bind to fibronectin (FN). The authors employed properly a set of powerful techniques such as HPLC-MS, MALDI-TOF MS and DPI to analyze the effects of molecular weight and hydroxylation modification of collagen peptides on FN binding activity. Very interesting paper, easy-going and methods, as well the results, were clearly presented. I recommend the approval of the paper, after some minor corrections described below:
- abstract, line 12 - Which relevant biological interactions? Be more especific, since it will justify the study.
- Lines 59 - 61 - I really do not understand this sentence. Maybe it will be interesting to rewrite it.
- Lines 243-244 - I think this sentence should belong to Fig. 5 legend.
- Table 1 - What does it mean the residues coloured in red and green? The text explanation should be in the table description.
- Figs. 9, 10 and 11 - Please, choose another visible color for the TE lines.
- Table 2 - six synthesized peptides? Should be four peptides.
- Acknowledgments are incomplete.
Author Response
- abstract, line 12 - Which relevant biological interactions? Be more especific, since it will justify the study.
Response: As reviewer's advice, we polished the abstract and shorten the content within 200 words.
- Lines 59 - 61 - I really do not understand this sentence. Maybe it will be interesting to rewrite it.
Response: We have rewritten this sentence to make it clearer. We hope you could understand what we are trying to express now.
- Lines 243-244 - I think this sentence should belong to Fig. 5 legend.
Response: Thank you for your reminding. We have revised this in the manuscript.
- Table 1 - What does it mean the residues coloured in red and green? The text explanation should be in the table description.
Response: Thank you for your kind suggestion. We have added the text explanation in the table note. However, as for the green highlighted text, we have deleted the highlighted mark. We think “GI” maybe also a specific sequence, but we can’t draw this conclusion at this time. In the future, we will go on the relative study.
- Figs. 9, 10 and 11 - Please, choose another visible color for the TE lines.
Response: We have revised the color for TE lines in Figure 9, 10 and 11. Now the lines are shown in red color.
- Table 2 - six synthesized peptides? Should be four peptides.
Response: We really appreciate your kind help to improve our manuscript. We have revised this mistake.
- Acknowledgments are incomplete.
Response: Thank you for your kind reminding. We have completed the acknowledgments. All the related researchers are listed in the author list. Therefore, we have “Not applicable” for the Acknowledgments.

Reviewer 2 Report
The manuscript titled “Identification of Collagen Peptides with Fibronectin-Binding 2 Properties and Functional Evaluation” by Liu et al. is a research paper mainly investigated the binding sites of collagen peptides with fibronectin. They found that collagen peptides with FN binding properties all contain GPAG and GPPG sequences. Moreover, they found that the binding ability of collagen peptides with FN are highly dependent on the molecular weight and hydroxylation modification of the peptides. The objective of this research is of significance and has not been well studied previously. This manuscript may provide important insight to fill this gap by conducting a systematic investigation. However, the overall flow needs to be improved. The poor written also reduces my enthusiasm. A major revision is recommended.
1. Language problems, please check the manuscript thoroughly.
e.g. Line 16 the molecular weight of these peptides was mainly concentrated….
Line 30 Using hydroxylation modification on determined by MALDI-TOF….
Line 42 what is denatured primarily type I collagen?
Line 59-61 attributes to contributes to
in collagen than in nature collagen?
Lines 184-185 ?
Line 186 what buffer?
2. The abstract needs to be more concise. The descriptions on how molecular weight of collagen peptides affect the binding activity with FN are extremely tedious. Please cut the abstract to around 200 words per the author instructions of the journal.
3. In the abstract, the statements in lines 16-17 and 22-24 confused me. According to the DPI results, peptides with molecular weights ranging from 2 to 30 kDa demonstrated better binding with FN. Why is there a mention on those between 1 and 2 kDa? Please make it clearer.
4. Lines 57-59, I don’t understand why there is a causal relationship between the two sentences. Hydroxyproline is important to collagen-FN binding because it is a unique amino acid of collagen? Then why does gelatin show better binding? Or hydroxyproline is important to collagen-FN binding because it is age and gender-dependent? What is the logic here?
5. Following this sentence, what is the justification by saying that molecular weight plays a role as gelatin shows better binding with FN compared to collagen?
6. The authors highlighted collagen in the title, abstract, and introduction. But why was bovine gelatin mainly used instead in the experimental design? Please clarify and make some justification.
Author Response
- Language problems, please check the manuscript thoroughly.
e.g. Line 16 the molecular weight of these peptides was mainly concentrated….
Line 30 Using hydroxylation modification on determined by MALDI-TOF….
Line 42 what is denatured primarily type I collagen?
Line 59-61 attributes to contributes to
in collagen than in nature collagen?
Lines 184-185 ?
Response: We have rewritten this sentence to make it clearer.
Line 186 what buffer?
Response: We have added the information of the buffer.
Response: We really appreciate your kind help to improve our manuscript. We have revised the above language problems the reviewer referred to. At the same, we have checked our manuscript thoroughly to improve our paper and try our best to make our paper easy to understand for all the readers.
- The abstract needs to be more concise. The descriptions on how molecular weight of collagen peptides affect the binding activity with FN are extremely tedious. Please cut the abstract to around 200 words per the author instructions of the journal.
Response: Thank you for your kind suggestion. We have revised our abstract. The total number of the abstract is 171 words now.
- In the abstract, the statements in lines 16-17 and 22-24 confused me. According to the DPI results, peptides with molecular weights ranging from 2 to 30 kDa demonstrated better binding with FN. Why is there a mention on those between 1 and 2 kDa? Please make it clearer.
Response: The sentence describing the molecular weight in lines 16-17 and 22-24 didn’t give valuable information on binding properties. We have deleted this sentence.
- Lines 57-59, I don’t understand why there is a causal relationship between the two sentences. Hydroxyproline is important to collagen-FN binding because it is a unique amino acid of collagen? Then why does gelatin show better binding? Or hydroxyproline is important to collagen-FN binding because it is age and gender-dependent? What is the logic here?
Response: Hydroxyproline content in collagen is age and gender-dependent, affecting collagen-FN binding process and further regulating cell migration. However, it is not clear that hydroxylation position is FN-binding associated. Thus, we synthesized two peptides with different hydroxylation position. The result indicated that not all hydroxylation of hydroxyproline is FN-binding associated.
- Following this sentence, what is the justification by saying that molecular weight plays a role as gelatin shows better binding with FN compared to collagen?
Response: The molecular weight of peptides in gelatin shows important influence on FN-binding properties. For peptides containing FN-binding domain, low MW peptide show low molecular flexibility and diffusional resistance in solution.
- The authors highlighted collagen in the title, abstract, and introduction. But why was bovine gelatin mainly used instead in the experimental design? Please clarify and make some justification.
Response: We revised the description on collagen/gelatin in the manuscript. Thanks a lot for your kind reminding.

Round 2
Reviewer 2 Report
My concerns have been well addressed. The revised version is acceptable for publication to me.